# Resveratrol and 3,3′-Diindolylmethane Differentially Regulate Aryl Hydrocarbon Receptor and Estrogen Receptor Alpha Activity through Multiple Transcriptomic Targets in MCF-7 Human Breast Cancer Cells

**DOI:** 10.3390/ijms241914578

**Published:** 2023-09-26

**Authors:** Siddhartha Das, Venkata S. Somisetty, Stine M. Ulven, Jason Matthews

**Affiliations:** 1Department of Nutrition, Institute of Basic Medical Sciences, University of Oslo, 0317 Oslo, Norway; siddhartha.das@medisin.uio.no (S.D.); v.s.somisetty@medisin.uio.no (V.S.S.); smulven@medisin.uio.no (S.M.U.); 2Department of Pharmacology and Toxicology, University of Toronto, Toronto, ON M5S 1A8, Canada

**Keywords:** AHR, ERα, diindolylmethane, resveratrol, ChIP sequencing, RNA sequencing

## Abstract

Inhibitory crosstalk between estrogen receptor alpha (ERα) and aryl hydrocarbon receptor (AHR) regulates 17β-estradiol (E2)-dependent breast cancer cell signaling. ERα and AHR are transcription factors activated by E2 and 2,3,7,8-tetrachlorodibenzo-p-dioxin (TCDD), respectively. Dietary ligands resveratrol (RES) and 3,3′diindolylmethane (DIM) also activate ERα while only DIM activates AHR and RES represses it. DIM and RES are reported to have anti-cancer and anti-inflammatory properties. Studies with genome-wide targets and AHR- and ERα-regulated genes after DIM and RES are unknown. We used chromatin immunoprecipitation with high-throughput sequencing and transcriptomics to study ERα as well as AHR coregulation in MCF-7 human breast cancer cells treated with DIM, RES, E2, or TCDD alone or E2+TCDD for 1 and 6 h, respectively. ERα bound sites after being DIM enriched for the AHR motif but not after E2 or RES while AHR bound sites after being DIM and E2+TCDD enriched for the ERE motif but not after TCDD. More than 90% of the differentially expressed genes closest to an AHR binding site after DIM or E2+TCDD also had an ERα site, and 60% of the coregulated genes between DIM and E2+TCDD were common. Collectively, our data show that RES and DIM differentially regulate multiple transcriptomic targets via ERα and ERα/AHR coactivity, respectively, which need to be considered to properly interpret their cellular and biological responses. These novel data also suggest that, when both receptors are activated, ERα dominates with preferential recruitment of AHR to ERα target genes.

## 1. Introduction

Breast cancer is the most common cancer in women and is a serious global health concern [1]. The growth of approximately 70% of breast cancers depends on estrogen (i.e., 17b-estradiol (E2)) and estrogen receptor (ER) signaling, with estrogen receptor α (ERα) playing a more prominent role in cancer progression than estrogen receptor β (ERβ) [2]. ERs are ligand-activated transcription factors and members of the nuclear receptor superfamily of transcription factors. They regulate gene expression in two ways, via direct DNA-binding to estrogen responsive elements (EREs) in regulatory regions of their target genes or via protein–protein interactions with other transcription factors, such as the aryl hydrocarbon receptor (AHR). Once bound to DNA, either directly via EREs or through tethering to other transcription factors, ERs recruit coregulator proteins resulting in changes in gene expression [3].

AHR is a ligand-dependent transcription factor and a member of the basic helix–loop–helix/PAS (Period- aryl hydrocarbon nuclear translocator (ARNT) Single-minded) family. AHR was initially identified as a key factor in mediating the toxicity of environmental contaminants, such as 2,3,7,8-tetrachlorodibenzo-p-dioxin (TCDD; dioxin), but it binds several other endogenous, dietary, and xenobiotic compounds. After binding the ligand, AHR translocates into the nucleus, heterodimerizes with ARNT, and binds to specific DNA sequences termed aryl hydrocarbon receptor response elements (AHREs). The activated AHR/ARNT heterodimer recruits coregulators leading to changes in target gene expression, including cytochrome P4501A1 (*CYP1A1*), cytochrome P4501B1 (*CYP1B1*), and TCDD-inducible poly-ADP-ribose polymerase (*TIPARP*; *PARP7*) [4,5,6].

Many research groups including our own have reported reciprocal crosstalk between ERα and AHR [7,8]. Activated AHR inhibits many ER-dependent responses; because of this, AHR agonists have been suggested as potential therapeutic targets for ERα+ breast cancer [9]. In support of this, AHR positive breast cancer patients have a relatively better prognosis than those with AHR negative breast cancer [10]. Some of our own work has characterized the genome-wide overlapping binding profiles of AHR and ER, after activation with TCDD, which in many cases leads to repression of ERα responses [11]. ERα-dependent modulation of AHR-regulated transcription is less clear and has been reported to vary from activation to inhibition to having no effect [12]. These differential outcomes may be due to cell context and the specific responses examined. Much of our understanding of the crosstalk between ERα and AHR is after treatment with their respective prototypical ligands, E2 and TCDD, respectively. However, both ERα and AHR are activated/inhibited by numerous compounds including a wide range of dietary phytochemicals, such as resveratrol and 3,3′diindolylmethane (DIM).

Resveratrol (3,4′,5-trihydroxystilbene; RES) is a polyphenol present in grapes, red wine, berries, and nuts [13] that has attracted a lot of research attention because it acts as an antioxidant, an anti-inflammatory agent, and a pro-apoptotic agent. These actions are thought to contribute to RES’s anti-carcinogenic, cardioprotective, and anti-aging properties [14]. In addition, RES is a potent inhibitor of some phase I drug metabolizing enzymes such as CYP1A1, CYP1B1, and CYP1A2 [15]. RES regulates the activity of many proteins including sirtuin 1, cyclooxygenase enzymes, peroxisome proliferator-activated receptor gamma, and AHR [16,17]. RES is an AHR antagonist and inhibits AHR-dependent transcription by preventing AHR/ARNT binding to the AHRE [18]. In contrast, RES is a phytoestrogen and activates ERα signaling [19]; although, there are some studies reporting that RES inhibits ERα activity [20]. The ability of RES to inhibit signaling may be, in part, due to its ability to inhibit aromatase activity [21].

Indole-3-carbinol (I3C) is a phytochemical and metabolic product of glucobrassicin, which is found in cruciferous vegetables such as broccoli, kale, and brussels sprouts. Upon ingestion, I3C is metabolized in the stomach to the AHR activators indolo [3,2-b]carbazole (ICZ) and DIM [22]. DIM has anti-cancer and anti-inflammatory properties and has been studied as a dietary supplement for breast cancer prevention and treatment. DIM has been shown to have both estrogenic and anti-estrogenic effects some of which are AHR mediated. In some studies, DIM has been shown to activate ERα and stimulate expression of estrogen-responsive genes which require protein kinase A signaling. High doses of 50 mM DIM have antiproliferative effects in most cancer cell lines [23,24]. DIM inhibits aromatase/cytochrome P45019A1 (CYP19A1) activity, which may contribute to its anti-cancer effects in estrogen-dependent breast cancers [25]. DIM also activates AHR causing increased nuclear translocation of AHR and subsequent increased expression of AHR target genes, such as CYP1A1 [26]. 

Our findings suggest that dietary intake of RES and DIM will modulate both receptors’ signaling pathways. However, there are no studies to look at the genome-wide binding profiles of ERα and AHR as well as the associated gene expression changes after exposure to RES and DIM. To fill this knowledge gap, we treated human breast adenocarcinoma cells, MCF-7, with DIM, RES, E2, or TCDD alone or in combination and determined the ERα- and AHR-binding and gene expression profiles and using chromatin immunoprecipitation (ChIP) sequencing and RNA sequencing, respectively.

## 2. Results

### 2.1. Comparison of ERα-Binding Profiles after Treatment with E2, RES, and DIM

To determine the genome-wide ERα-binding profiles of DIM and RES compared with those of E2, ChIP-sequencing was performed on extracts from MCF-7 cells treated with 10 nM E2, 10 μM RES, 10 μM DIM, or DMSO (solvent control) for 1 h. MCF-7 cells are E2-responsive and are often used in vitro to study estrogen receptor positive breast cancers. In addition, MCF-7 cells also express AHR, and they are routinely used to study ERα and AHR crosstalk [11,12]. A dose of 10 nM E2 is commonly used for in vitro studies with human cell lines and induces a maximal response when measuring ERα target genes like trefoil factor [18,27]. The RES and DIM doses of 10 μM were chosen based on our previous work studying their ability to activate ERα [18]. Cells were treated for a 1 h time point because previous time course ChIP studies revealed that AHR and ERα recruitment peaks between 45 and 90 min [8]. We first determined the ERα-bound regions in the absence of ligand, by determining the ERα-bound peaks in the DMSO treated compared with the immunoglobulin G (IgG) immunoprecipitated samples as described in Materials and Methods. Using this approach, we identified 23,793 ERα-bound sites after DMSO treatment. These findings support previous studies that ERα exhibits significant constitutive chromatin binding in the absence of external ligand activation [28]. To determine the ERα-binding profiles after ligand activation, we compared MCF-7 cells treated for 1 h with 10 nM E2, 10 μM RES, or 10 μM DIM to the DMSO-treated samples. We found that E2, RES, and DIM resulted in 48,753, 35,479, and 31,482 ERα-bound regions, respectively. Overlap analysis of these ERα-bound regions revealed that there were 31,090 (88%) and 26,839 (86%) of RES–ERα- and DIM–ERα-bound regions that were also occupied by ERα after E2 (Figure 1A,B). When we overlapped the ERα-bound regions after RES and DIM, we found that DIM–ERα sites had fewer unique ERα-binding regions at 8134 compared to 12,193 for RES (Figure 1C).

Annotating the peaks to their nearest gene revealed that E2–ERα, RES–ERα, and DIM–ERα sites mapped to 17,592, 15,831, and 14,638 genes, respectively. Overlap analysis showed that 13,289 (91%) of the DIM–ERα and 14,347 (91%) of the RES–ERα genes were common with E2–ERα genes (Figure 1D,E). There were 12,449 genes shared between RES– ERα and DIM–ERα sites, which corresponded to 79% and 84% of their closest genes, respectively (Figure 1F).

We next annotated the binding sites with respect to genomic location and found that ERα-bound regions after E2 and DIM contained a similar percentage of promoter containing sequences at 16.8%, while RES treatment also resulted in a slightly lower percentage of promoter-containing regions at 14.8% (Figure 1G–I). These data were also confirmed by plotting the distribution of the distance of the binding sites relative to the transcription start site (Figure 1J–L). Overall, the ChIP-sequencing data show that E2, RES, and DIM induce similar genomic ERα-binding profiles with a high degree of overlapping binding sites and corresponding closest genes. 

To gain more insight into the transcription factors that might co-regulate ERα-bound genes, we did motif analysis using the top 1000 ligand-enriched sites for ERα binding after E2, RES, or DIM treatment. As expected, the ERE was the most enriched motif for all the three ligands (Table 1). However, DIM–ERα-bound regions were also enriched for an AHRE which agrees with its ability to activate AHR and suggests that ERα and AHR are co-recruited and co-regulate many target genes [11]. In contrast, RES is an AHR antagonist and prevents AHR binding to AHREs [18]. In line with this, we did not observe enrichment of AHREs in RES–ERα-bound regions.

### 2.2. Transcriptomic Changes after E2, RES, or DIM Regulated by ERα

To determine whether the ligand-induced ERα-bound regions corresponded with changes in gene expression, we did RNA-sequencing on extracts from MCF-7 cells treated for 6 h with DMSO, E2, RES, or DIM. We chose a 6 h treatment for the RNA sequencing because we were interested in primary target genes of ERα and AHR rather than secondary, downstream-activated genes. Differential expression analysis relative to DMSO revealed that treatment with E2, RES, and DIM resulted in 866 (518 up and 348 down), 577 (50 up and 227 down), and 446 (295 up and 151 down) differentially expressed genes at adjusted *p* value less than 0.01 and absolute fold change greater than 1, respectively (Appendix A). 

We found that 331 genes were significantly and commonly altered in all three treatments compared with DMSO (Figure 2A). Of these 331 genes, many known E2-responsive genes were identified, including growth-regulating estrogen receptor binding 1 (*GREB1*), carbonic anhydrase 12 (*CA12*) and progesterone receptor (*PGR*). For E2 and RES, an additional 172 differentially expressed genes were common to both treatments, revealing that 503 (87%) of the 577 differentially expressed genes identified after RES treatment were also found after E2 treatment. For E2 and DIM, an additional 52 differentially expressed genes were common in both treatments, revealing that 383 (86%) of the differentially expressed genes identified after DIM treatment were also found after E2 treatment. However, E2 treatment also resulted in 311 differentially expressed genes that were unique to E2 only. Consistent with the ability of DIM to act as an AHR agonist, *CYP1A1* levels were increased after DIM treatment but decreased after E2 and RES. 

We next identified genes that were closest to an ERα-binding site after various ligand treatments using ChIP sequencing and are differentially expressed from the RNA sequencing for the corresponding ligand (Appendix A). We found 532 (339 up, 193 down) of the 866 E2 differentially expressed genes were also closest to an ERα-binding site, which included known ERα targets *GREB1*, *CA12*, and *PGR* (Figure 2B,C). For RES and DIM, 393 (244 up, 149 down) and 278 (200 up, 78 down) differentially expressed genes were closest to an ERα-binding site, respectively. Like that observed for E2, GREB1, CA12, and PGR were also upregulated (Figure 2D). However, in the ERα-regulated genes after DIM treatment, we observed that the AHR-regulated gene, *CYP1A1*, was one of the top upregulated genes (Figure 2E). In contrast, *CYP1A1* was downregulated in both the E2–ERα and RES–ERα gene sets, suggesting that ERα differentially regulates AHR target genes depending on the nature of the ligand.

Ontology analysis for the ERα-regulated genes indicated that 25 of the 34 (73.5%) significantly enriched gene ontology terms for RES–ERα differentially expressed genes, while only 24 of the 46 (52%) terms enriched for DIM–ERα differentially expressed genes were also among E2–ERα differentially expressed genes, respectively (Appendix A).

This suggests that DIM–ERα regulates several different biological functions than E2– ERα. One possible explanation for this is that DIM also activates AHR, supported by the observation that DIM–ERα sites were enriched for AHR motif while E2–ERα- and RES–ERα-bound sites did not (Table 1).

### 2.3. Comparison of AHR-Binding Profiles after Treatment with TCDD, E2+TCDD, RES, or DIM

Because of the differential abilities of DIM and RES to regulate AHR and the observed binding of ERα to AHR motifs after DIM treatment, we determined the genome-wide binding profiles of AHR ChIP-sequencing after a one-hour treatment with DMSO (solvent control), 10 nM TCDD, 10 μM RES, 10 μM DIM, or 10 nM E2 and 10 nM TCDD co-treatment (E2+TCDD). A dose of 10 nM TCDD is routinely used as a maximal dose for studying AHR activity in human cell lines in vitro, as measured by the induction of *CYP1A1* mRNA levels. A dose of 10 μM RES effectively activates ERα but represses AHR signaling [18,29], while a dose of 10 μM DIM induces the recruitment of AHR and ERα to their target genes and the levels of their gene products [27]. The AHR-bound regions in the absence of a ligand stimulation were determined by comparing the DMSO–AHR-bound peaks against IgG-immunoprecipitated samples as described for ERα-bound regions. Using this approach, we identified 1726 AHR-bound sites after DMSO treatment. These findings support previous studies suggesting that AHR exhibits significant constitutive chromatin binding in the absence of external ligand activation as previously reported [30]. We next determined the AHR peaks in the ligand treated samples relative to DMSO. We found that TCDD, DIM, and E2+TCDD treatment resulted in 7762, 14,737, and 9468 AHR-bound regions, respectively. In agreement with RES acting as an AHR antagonist, we only observed four AHR-bound regions after RES treatment. Overlapping of the AHR-binding sites revealed that 7776 (52.8%) and 4513 (47.7%) of the AHR-bound sites after DIM and E2+TCDD were unique, suggesting AHR acquires new targets for regulation after DIM and E2+TCDD treatments that were not seen after TCDD treatment (Figure 3A,B). We observed that 8941 (95%) of the E2+TCDD–AHR-bound regions overlapped with those of DIM–AHR, suggesting the E2+TCDD treatment results in AHR’s binding to most of the genes targeted for regulation after DIM (Figure 3C).

Annotating the peaks to their nearest gene revealed that the AHR-binding sites after TCDD, DIM, or E2+TCDD treatments were closest to 5255, 8488, and 5864 genes, respectively. Overlap analysis revealed that 3604 and 2090 genes closest to an AHR binding site after DIM and E2+TCDD treatment were unique and not observed after TCDD which were 42.5% and 35.6% of the total genes targeted for regulation after each treatment, respectively (Figure 3D,E). Moreover, 5664 (96%) of the closest genes to E2+TCDD–AHR-bound regions were common with those of DIM–AHR (Figure 3F). Annotating the AHR-binding sites with respect to their genomic location revealed that 16.44% of the AHR-binding sites after TCDD were promoter sites while 18.95% and 19.63% of the AHR binding sites after E2+TCDD and DIM were promoter sites (Figure 3G–I). We also confirmed the promoter distribution of the AHR-binding sites by plotting the distribution of the distance of the AHR-binding sites relative to the transcription start site after TCDD, E2+TCDD, and DIM (Figure 3J–L). Overall, the AHR ChIP-sequencing data indicate that DIM and E2+TCDD induce more AHR-binding events compared with TCDD and suggest that AHR may be regulating new gene sets after concurrent AHR and ERα activation not seen after TCDD treatment.

### 2.4. Motif Analysis Using AHR Ligand-Enriched Sites after Treatment with TCDD, E2+TCDD, or DIM

To profile transcription factors targeted for regulation by AHR after E2+TCDD and DIM relative to TCDD, we did motif analysis using the top 1000 AHR ligand-induced binding sites (Table 2, Appendix A). As expected, we found that an AHRE was the highest enriched motif after TCDD treatment. However, for both DIM–AHR and E2+TCDD–AHR, an ERE motif was the most significantly enriched motif, with an AHRE motif being the second highest enriched motif for both conditions. Nuclear receptor subfamily 1, group A, member 2 (NR1A2) and nuclear receptor subfamily 2, group F, member 2 (NR2F2) motifs were among the top five most enriched motifs after E2+TCDD treatment but not after DIM treatment. Of note, the NR1A2 and NR2F2 motifs were among the top five most enriched motifs for E2–ERα sites.

We next did motif analysis using the AHR and ERα co-occupied regions with the top 1000 peaks of ERα and AHR after E2+TCDD and DIM to find transcription factor motif sequences regulated by both which may not have been detected using enriched sites of each receptor individually. We found that co-occupied sites after both DIM and E2+TCDD enrichment for AHRE included neuronal PAS domain protein 4 (NPAS4), ERE, transcription factor AP-2 gamma (AP-2gamma), and hypoxia-inducible factor 1 beta (HIF1B) transcription factor motifs within the top five motifs, but the rankings were different (Table 3). Uniquely enriched transcription factor motifs among DIM co-occupied sites included transcription factor CP2-like 1 (TCFCP2L1); TGFB-induced factor homeobox 1 (TGIF1); GATA binding protein 3 (GATA3); nuclear receptor subfamily 4, group A, member 1 (NR4A1); and forkhead box K2 (FOXK2) among others, and neurofibromin 1 (NF1) and peroxisome proliferator activated receptor alpha (PPARA) were among E2+TCDD unique motifs (Appendix A). Collectively these data suggest that, although most of the E2+TCDD–AHR-bound genes were like those of DIM–AHR, there are also unique transcription factor motifs enriched by AHR and ERα under the different ligand treatments. 

### 2.5. Comparison of Gene Expression Changes Induced after AHR Activation Compared to Coactivation of AHR and ERα via E2+TCDD and DIM

We next determined and compared the transcriptomic profiles after treatment with TCDD, DIM, and E2+TCDD. The TCDD treatment only resulted in 64 (57 up, 7 down) differentially expressed genes while we observed 665 differentially expressed genes (394 up, 271 down) after the E2+TCDD treatment compared to 446 after DIM (Appendix A). The largest subset of genes was the 323 differentially expressed genes after only E2+TCDD.

However, 311 genes were differentially expressed after both E2+TCDD and DIM treatments (Figure 4A). Of the 665 differentially expressed genes after E2+TCDD, 506 (76%) were also differentially expressed after E2 (Appendix A). Of the 405 E2+TCDD-bound ERα differentially expressed genes, 296 (73%) were the same as those regulated by E2-ERα. ERα was also recruited to the AHR target gene, *CYP1A1*, after E2+TCDD and DIM.

To find genes that are direct targets of AHR regulation, we focused on differentially expressed genes closest to an AHR-binding site after TCDD, DIM, or E2+TCDD treatment. The largest subsets of genes were those which had both an AHR- and ERα-binding site closest to it after both E2+TCDD and DIM (Figure 4B). We also found that 40 out of the 64 TCDD–AHR differentially expressed genes and 222 of the 446 DIM–AHR differentially expressed genes included the known AHR target genes: *CYP1A1*, *CYP1B1*, and *TIPARP* (also known as *PARP7*) for TCDD and only *CYP1A1* and *CYP1B1* for DIM (Figure 4C,D). For E2+TCDD, 211 of the 665 differentially expressed genes were bound by AHR (Figure 4E). Among the genes closest to an AHR-binding site after E2+TCDD treatment, we found the AHR target genes of CYP1A1 and CYP1B1 but also the E2 target genes *GREB1*, *CA12*, and *PGR*, which were also observed after DIM treatment. Because the AHR bound closest to ERα-regulated genes after E2+TCDD and DIM and both DIM- and E2+TCDD-bound AHR sites were enriched for an ERE, we overlapped the ERα- and AHR-bound sites after E2+TCDD or DIM to find coregulated sites and genes. We found that 8904 or 90.5% of the AHR-bound sites overlapped with ERα-bound ones after E2+TCDD treatment, while 10,546 or 72% of the DIM–AHR-bound sites overlapped with those of ERα–DIM. This indicates that AHR and ERα might coregulate several genes after E2+TCDD or DIM treatment. Overlapping the AHR or ERα-bound sites differentially expressed genes after either DIM or E2+TCDD revealed that 204 of the 222 (91.9%) differentially expressed genes after DIM and 208 of the 211 (98.6%) (Appendix A) differentially expressed genes after E2+TCDD closest to an AHR-binding site also had an ERα-binding site (Figure 5A,B,D,E). Overlapping these differentially expressed genes revealed that 123 (60%) of the differentially expressed genes were the same for both treatments (Figure 5C). Enrichment analysis using these coregulated genes indicated that there were 30 significant gene ontology (GO) terms for the DIM-coregulated genes and 26 significant terms for the E2+TCDD-coregulated genes, with 9 common to both treatments (Table 4 and Appendix A). These data suggest that, when both ERα and AHR are activated, ERα dominants causing the preferential recruitment of AHR to ERα target genes.

## 3. Discussion

RES and DIM are phytochemicals that have received considerable attention due to their beneficial health effects and anti-cancer properties. Both compounds modulate the activities of several signaling pathways including ERα- and AHR-dependent responses [22,31,32]. RES activates ERα but inhibits AHR, while DIM activates both ERα and AHR. ERα–AHR crosstalk is well established, but most studies have characterized the signaling interplay between these two receptors after treatment with E2 to activate ERα or toxic chemicals, such as TCDD or 3-metylcholanthrene (3MC), to activate AHR [7,12]. Moreover, there are few reports of the impact of RES or DIM on ERα–AHR crosstalk at the genome-wide level. In this study, we used ChIP-sequencing and RNA-sequencing to determine how RES and DIM differentials affect ERα and AHR in MCF-7 breast cancer cells. Our data highlight that single ligand exposure can activate a diverse signaling response, which should be considered when assessing cellular responses. Moreover, our data suggest that, when both ERα and AHR are activated, ERα dominants causing the preferential recruitment of AHR to ERα target genes. Humans are exposed to these compounds through their diets and nutritional supplements, and their differential modulation of ERα and AHR signaling may have physiological consequences.

RES has beneficial effects on aging, inflammation, oxidative stress, and metabolism, which are proposed to be related to the activation of sirtuin 1 (SIRT1), protein kinase A, AMP-activated protein kinase, apoptosis, or autophagy [33]. RES has been shown to modulate inflammatory responses through its ability to activate ERα [29]. We observed that 87% of all differentially expressed genes induced by RES were also induced by E2, and 84.5% of the differentially expressed genes with an ERα-binding site after RES were also differentially expressed after E2 with an ERα-binding site among those closest to it, supporting an ERα-mediated mechanism of action for RES. Despite its estrogenic actions, RES exhibits antiproliferative effects in several cancer cell lines including breast, prostate, and colon cancers. The antiproliferative effects of RES are independent of ER as they have been reported for both ER-positive and ER-negative cancer cell lines [34]. In addition, RES has also been reported to inhibit ERα activity, suggesting that the actions of RES may be context- and cell line-specific [20].

Consistent with RES acting as an AHR antagonist, we only observed 4 AHR-bound regions after a 1 h treatment of MCF-7 cells compared with 7762 for TCDD and 14,737 for DIM. Reduced expression of AHR target genes, like *CYP1A1*, was also observed after RES treatment. Based on these findings, inhibition of AHR action must also be considered after short-term RES exposure. However, RES, like quercitin and curcumin, is a potent inhibitor of CYP1A1 enzymatic activity, an enzyme that metabolizes many dietary and endogenous ligands [35]. Altered metabolic degradation of endogenous ligands can lead to altered cellular responses and AHR activation. For example, overnight treatment with quercitin, curcumin, or RES indirectly activates AHR by inhibiting CYP1A1 and, thus, prevents the degradation of the endogenous AHR ligand 6-formylindolo [3,2-b]carbazole (FICZ) [35]. This was not evident in our study since we were interested in direct ERα or AHR target genes and limited our ligand exposure to 6 h.

DIM is one the most abundant and best characterized bioactive compounds found in commonly consumed cruciferous vegetables or as a supplement. DIM exhibits a vast range of pleiotropic anti-tumor effects and has been reported to modulate all stages of breast tumor development [36]. DIM orchestrates its actions through ERα, AHR, and other signaling pathways, including nuclear factor kappa B (NF-κB), nuclear factor erythroid 2-related factor 2 (NFE2F2), and mitogen-activated protein kinases (MAPK), to modulate cell cycle, gene expression, CYP450 enzyme levels, and apoptosis [37]. The biggest difference between RES and DIM responses in the MCF-7 cells was the activation of AHR and the induction of known AHR target genes like *CYP1A1*. Compared with TCDD alone, DIM resulted in more genome-wide AHR binding and differentially expressed genes that were also associated with an AHRE. Our data suggest that this is due to DIM’s ability to activate both ERα and AHR. This is supported by our observation that AHR binds to additional genomic sequences after a DIM or E2+TCDD treatment compared with TCDD alone. Moreover, over 95% of the E2+TCDD–AHR-bound regions were common with those of DIM–AHR. Activated AHR is reported to inhibit many ER-dependent responses; because of this, AHR agonists have been suggested as potential therapeutic targets for ERα+ breast cancer. AHR has been implicated in the ability of DIM to inhibit E2-induced proliferation of MCF-7 cells. We observed that genes from the receptor tyrosine kinase signaling were significantly downregulated after DIM treatment but enriched after E2. Tyrosine kinase-signaling genes, *EPHB3*, *SH3TC2*, and *GRB7*, which are associated with poor disease-free survival and promote cancer progression and invasion [38,39], were downregulated by DIM–AHR.

Alterations in the levels and/or activities of cytochrome P450 enzymes can profoundly affect estrogen metabolism and breast cancer progression. The estrogen metabolite 2-hydroxyestrone (2OHE1) is protective, while 4-hydroxyestrone (4OHE1) is associated with tumor formation and increased DNA adduct formation [40,41]. DIM-dependent induction of CYP1A1 promotes the formation of 2OHE1 rather than 4OHE1, which supports anti-inflammatory effects and protects against tumor formation [42].

ERα-dependent modulation of AHR-regulated transcription is complex and has been reported to vary from activation to inhibition to having no effect [12]. We observed 64 differentially expressed genes after TCDD treatment, 40 of which were bound by AHR. We observed 222 AHR-bound differentially expressed genes after DIM treatment, with 204 of them also bound by ERα. Only 14 of 204 genes were also differentially expressed after TCDD. This suggests that when both ERα and AHR are activated, ERα recruits AHR to its target genes. In support of this, we found that 96% of the genes bound by AHR after E2+TCDD were also bound by AHR after DIM. In both treatments, an ERE was the highest enriched motif followed by an AHRE.

Overall, our study has shed further light on how ERα and AHR activities are modulated by the dietary ligands RES and DIM, as well as how many dietary and exogenous mediate their actions through multiple signaling cascades rather than a single cellular target. These data further highlight the complex crosstalk between ERα and AHR and suggest that, when both receptors are activated, ERα dominants causing the preferential recruitment of AHR to ERα target genes. Although our study focused on ERα and AHR, only 62% and 68% of the differentially expressed genes after DIM and RES are direct ERα or AHR targets suggesting there are other signaling pathways regulating gene expression changes after treatment with these ligands. Future “omics” studies are warranted to uncover if some of the genes found to not be directly regulated by ERα in our study are targets of those transcription factors.

## 4. Materials and Methods

### 4.1. Chemicals and Antibodies

DMSO, DIM, E2, and RES were from Merck (Darmstadt, Germany). The 2,3,7,8-tetrachlorodibenzo-p-dioxins (TCDD) were purchased from Accustandard (New Haven, CT, USA). Antibodies used for ChIP-sequencing include anti-AHR (H-211; Santa Cruz Biotechnology, Dallas, TX, USA) and normal rabbit immunoglobin (sc-2027; Santa Cruz Biotechnology). Protein A Agarose Fast Flow beads from ThermoFisher Scientific (Waltham, MA, USA) were used for all ChIP-sequencing experiments. All other reagents were from Merck unless stated otherwise.

### 4.2. Cell Culture

MCF-7 HTB-22 human breast cancer cells were obtained from ATCC (Manassas, VA, USA). They were cultured in low-glucose (1 g/L) DMEM (Dulbecco’s modified Eagle’s media) supplemented with 1% (*v*/*v*) penicillin/streptomycin (P/S) either 10% (*v*/*v*) fetal bovine serum (FBS) or 5% dextran-coated, charcoal-treated FBS. Cell culture media and supplements were purchased from Merck Life Science (Rahway, NJ, USA).

### 4.3. ChIP-Sequencing and Data Analysis

MCF-7 cells were dosed for 1 h with 10 nM or 10 µM of various ligands or DMSO and the ChIP-sequencing experiment was performed in triplicates (GSE232235). The Illumina raw FASTQ files were mapped to the human genome assembly hg38 using Bowtie2 (v2.4.4.) [43]. The resulting SAM files were converted to BAM files, sorted, and indexed using samtools (v1.14.) [44]. For peak calling, the replicates were pooled using samtools. Peak calling was performed using MACS2 (v3.0.0.) with default settings [45]. To identify constitutively activated AHR- and ERα-binding sites after ligand or DMSO treatment, we used a peak calling approach where IgG-treated samples were considered as the background and the DMSO or ligand treated samples as the treatment. For identifying ligand-induced peaks, the pooled BAM files for a specific a ligand treatment was compared against that of DMSO using MACS2. The resulting BED file containing all the peak regions that passed the q value cutoff of 0.05 were considered as peaks. High risk genomic regions with high ChIP signals such as centromeres, telomeres, and satellite repeats were removed using the ENCODE consortium blacklisted regions and BEDTools (v2.18.) [46]. All binding regions were overlapped using ChIPpeakAnno R package [47]. The annotation of the peaks to their nearest regions was also carried out using the ChIPpeakAnno (v3.32.0.) R package. The Hypergeometric Optimization of Motif Enrichment (HOMER) (v4.11.1.) analysis suite was used for de novo motif discovery [48]. The HOMER-identified motifs were compared to known motifs in the JASPAR database to identify known motifs bound by AHR and ERα after ligand treatment [49].

### 4.4. RNA Sequencing and Data Analysis

The MCF-7 cells were treated for 6 h with 10 nM or 10 µM of various ligands or DMSO in triplicate. Total RNA was isolated with the Aurum™ total RNA mini kit from BioRad (Hercules, CA, USA), following the manufacturer’s protocol. The RNA yield was assessed with NanoDrop™ 1000 (Thermo Fisher Scientific, Waltham, MA, USA). The RNA quality was evaluated using the Agilent 2100 bioanalyzer from Agilent (Palo Alto, CA, USA) with the RNA 6000 Nano LabChip kit and subjected to RNA sequencing. They were multiplexed and sequenced using strand-specific TruSeq RNA-sequencing library prep as single end reads on a NextSeq 500 (Illumina, San Diego, CA, USA) machine using 75 base reads.

The reads were mapped to the human reference transcriptome assembly (hg38) using Hisat2 (v2.2.1.) to generate BAM file [50]. The BAM file and the hg38 GTF file from UCSC was supplied to htseq-count (v0.13.5.) to generate raw count of reads aligning to the transcriptome with the flags “-s reverse”, “-m union”, and “-t exon” [51]. Differentially expressed genes were determined using DESeq2 (v1.38.3.) by supplying the raw read count to it [52]. Only genes that had counts per million (cpm) greater than 1 in at least 2 samples were used for the differential expression. *p* values from the differential expression were corrected using Benjamin Hochberg FDR correction and only those genes with adjusted *p* values less than 0.01 and absolute log fold change greater than 1 were considered as significantly differentially expressed. Each experiment was performed in triplicate.

### 4.5. Enrichment Analysis for Differentially Expressed Genes

Genes that were significantly up- and downregulated after ligand treatment and were close to an ERα- or AHR-binding site after ligand treatment or coregulated by AHR and ERα were subjected to gene ontology analysis using “enricher” function from cluster Profiler (v4.6.2.) R package [53]. Briefly, a total of 20,493 genes qualified for the expression cutoff of counts per million mapped reads greater than 1 in at least 2 samples and were used as background for the enrichment analyses. The Gene Ontology library from the “msigdbr” (v7.5.1.) R package was obtained by specifying species as “*Homo sapiens*” [54]. These data have enrichment information from multiple different databases. We filtered them to use the GO terms only. The “enricher” function uses a hypergeometric test to find GO terms overrepresented among the significant genes using the Msig database GO terms. Briefly, the significantly altered genes from RNA sequencing were used as genesets of query to “enricher”, and an FDR adjusted *p* value cutoff of 0.01 was used to detect significantly enriched terms after correcting for multiple testing.

## 5. Conclusions

In this study, we found that RES and DIM differentially regulate ERα and AHR action and likely other signaling pathways via multiple transcriptomic targets, which need to be considered to properly interpret their cellular and biological responses. These novel data further highlight the complex crosstalk between ERα and AHR and suggest that when both receptors are activated, ERα dominants causing the preferential recruitment of AHR to ERα target genes.

## Figures and Tables

**Figure 1 ijms-24-14578-f001:**
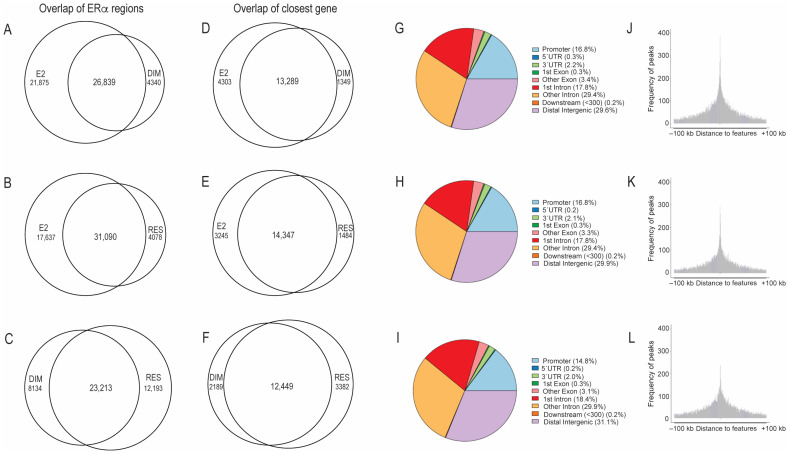
Characterization of ERα ligand induced binding sites in MCF-7 breast cancer cells after E2, DIM and RES treatment relative to DMSO. Overlap of E2, RES and DIM ligand induced ERα peaks (**A**–**C**). Overlap of E2, RES and DIM ligand induced ERα peaks nearest genes (**D**–**F**). Genome wide distribution of ERα ligand induced binding sites after E2, DIM and RES (**G**–**I**). Distribution of distance of the ligand induced binding sites relative to nearest gene transcription start site after E2, DIM and RES (**J**–**L**).

**Figure 2 ijms-24-14578-f002:**
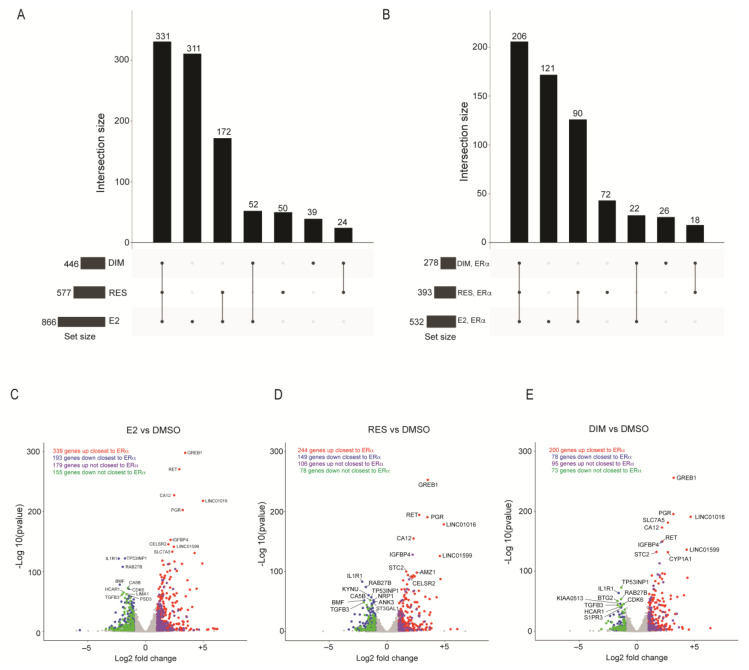
Ligand-induced gene expression changes using RNA sequencing data after treatment of MCF-7 breast cancer cells with E2, DIM, and RES. Upset plot for overlap of differentially expressed genes after E2, DIM, and RES (**A**). Upset plot for subset of differentially expressed genes after E2, DIM and RES closest to an ERα site only (**B**). Volcano plot for gene expression changes after E2, DIM, and RES relative to DMSO, separated into up or downregulated genes as well as closest or not to ERα lig-and-induced sites (**C**–**E**).

**Figure 3 ijms-24-14578-f003:**
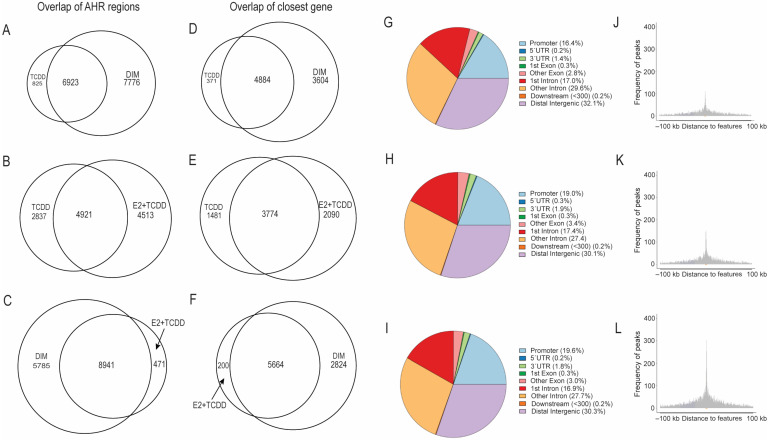
Characterization of AHR ligand-induced binding sites in MCF-7 breast cancer cells after TCDD, DIM, and E2+TCDD treatment relative to DMSO. Overlap of TCDD, DIM, and E2+TCDD ligand-induced AHR peaks (**A**–**C**). Overlap of TCDD, DIM, and E2+TCDD ligand-induced AHR peaks’ nearest genes (**D**–**F**). Genome-wide distribution of AHR ligand-induced binding sites TCDD, E2+TCDD, and DIM (**G**–**I**). Distribution of distance of the ligand-induced AHR-binding site relative to the nearest gene transcription start site after TCDD, E2+TCDD, and DIM (**J**–**L**).

**Figure 4 ijms-24-14578-f004:**
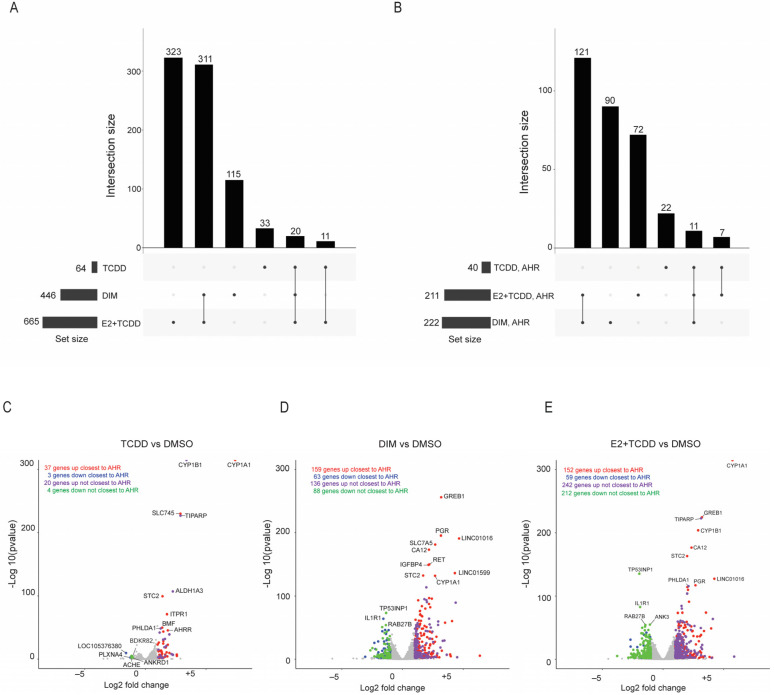
Ligand-induced gene expression changes using RNA-sequencing data after treatment of MCF-7 breast cancer cells with TCDD, DIM, and E2+TCDD. Upset plot for overlap of differentially expressed genes after TCDD, DIM, and E2+TCDD (**A**). Upset plot for differentially expressed genes after TCDD, DIM, and E2+TCDD closest to an AHR site only (**B**). Volcano plot for gene expression changes after TCDD, DIM, and E2+TCDD relative to DMSO, separated into up or downregulated genes as well as closest or not to AHR ligand-induced sites (**C**–**E**).

**Figure 5 ijms-24-14578-f005:**
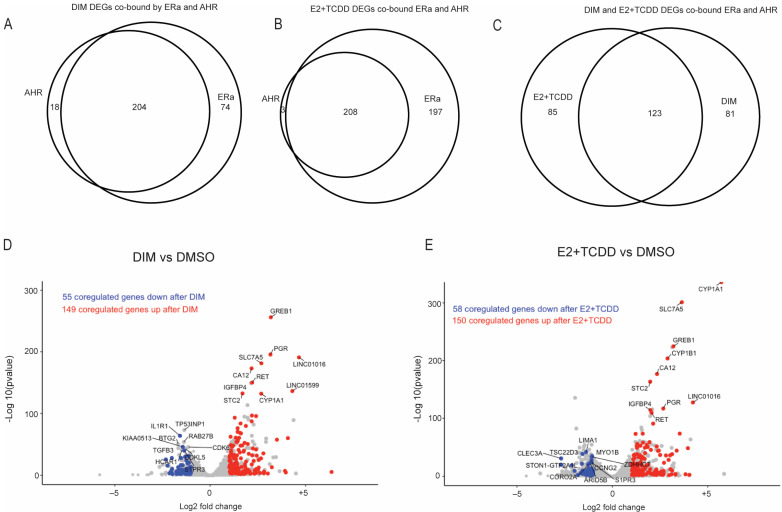
Coregulation by AHR and ERα after DIM and E2+TCDD treatments. Overlap of AHR- and ERα-nearest genes that are differentially expressed after DIM and E2+TCDD (**A**,**B**). Overlap of genes that are closest to both AHR and ERα after DIM or E2+TCDD (**C**). DIMSO vs. DIM and DMSO vs. E2+TCDD volcano plot with AHR- and ERα-coregulated genes highlighted (**D**,**E**).

**Table 1 ijms-24-14578-t001:** Motif analysis for top 1000 ERα ligand-induced sites after E2, DIM, and RES treatment.

	E2-Induced ERα-BoundRegions	DIM-Induced ERα-BoundRegions	RES-Induced ERα-BoundRegions
Rank	Motif(JASPAR Eval)	Motif(JASPAR Eval)	Motif(JASPAR Eval)
1	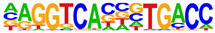 (ERE, 10^−224^)	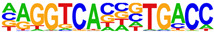 (ERE, 10^−155^)	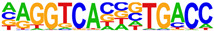 (ERE, 10^−225^)
2	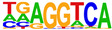 (NR1A1, 10^−92^)	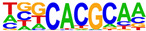 (AHRE, 10^−110^)	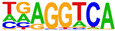 (NR1A2, 10^−105^)
3	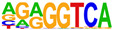 (NR2F2, 10^−81^)	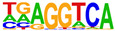 (NR1A2, 10^−69^)	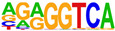 (NR2F2, 10^−77^)
4	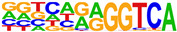 (NR2F6, 10^−60^)	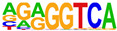 (NR2F2, 10^−57^)	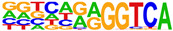 (NR2F6, 10^−68^)
5	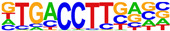 (ESRRG, 10^−52^)	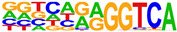 (NR2F6, 10^−39^)	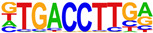 (ESRRB, 10^−58^)

**Table 2 ijms-24-14578-t002:** Motif analysis for top 1000 AHR ligand induced sites after TCDD, DIM, and E2+TCDD treatments.

	TCDD-Induced AHR-Bound Regions	DIM-Induced AHR-BoundRegions	E2+TCDD-Induced AHR-Bound Regions
Rank	Motif(JASPAR Match, Eval)	Motif(JASPAR Match, Eval)	Motif(JASPAR Match, Eval)
1	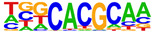 (AHRE, 10^−320^)	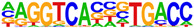 (ERE, 10^−160^)	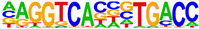 (ERE, 10^−199^)
2	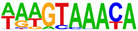 (MCF7-FOXA1, 10^−80^)	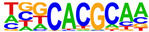 (AHRE, 10^−159^)	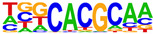 (NR1A2, 10^−84^)
3	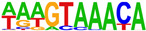 (LNCAP-FOXA1, 10^−77^)	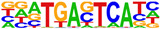 (FRA1, 10^−56^)	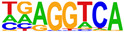 (NR1A2, 10^−70^)
4	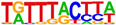 (FOXM1, 10^−72^)	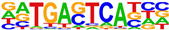 (FOSL2, 10^−51^)	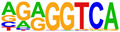 (NR2F2, 10^−45^)
5	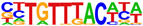 (FOXA2, 10^−67^)	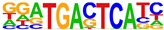 (FRA2, 10^−51^)	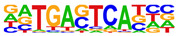 (FOSL2, 10^−41^)

**Table 3 ijms-24-14578-t003:** Motif analysis for top 1000 AHR and ERα ligand-induced sites overlapping regions after DIM and E2+TCDD treatment.

	AHR and ERα Overlapping Regions after E2+TCDD	AHR and ERα Overlapping Regions after DIM
Rank	Motif(JASPAR Best Match, Eval)	Motif(JASPAR Best Match, Eval)
1	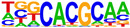 (AHRE, 10^−44^)	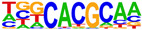 (AHRE, 10^−64^)
2	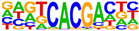 (Npas4, 10^−16^]	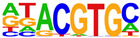 (HIF1B, 10^−17^)
3	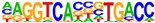 (ERE, 10^−13^)	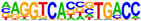 (ERE, 10^−15^)
4	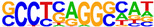 (AP-2gamma (AP2)/TFAP2C, 10^−12^)	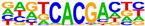 (Npas4, 10^−15^)
5	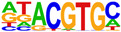 (HIF1B, 10^−10^)	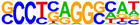 (AP-2gamma(AP2)/TFAP2C, 10^−12^)

**Table 4 ijms-24-14578-t004:** Enrichment terms common among coregulated genes differentially expressed after E2+TCDD and DIM.

Enrichment Term	Adj. *p* Value for E2+TCDD	Adj. *p* Value for DIM
External encapsulating structure	0.00561	0.00126
Microvillus membrane	0.00763	0.00126
Glycosaminoglycan binding	0.00716	0.00120
Skeletal system development	0.00132	0.00120
Endochondral bone morphogenesis	0.00762	0.00483
Tissue morphogenesis	0.00467	0.00483
Microvillus	0.00680	0.00734
Excretion	0.00833	0.00775
Response to corticosteroid	0.00845	0.00775

## Data Availability

The data are available in the GEO repository under the accession GSE232235.

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
