# Peer review of "Resveratrol and 3,3′-Diindolylmethane Differentially Regulate Aryl Hydrocarbon Receptor and Estrogen Receptor Alpha Activity through Multiple Transcriptomic Targets in MCF-7 Human Breast Cancer Cells"

_ijms, 2023, doi:10.3390/ijms241914578_

Round 1
Reviewer 1 Report
ijms-2607836
Title: Resveratrol and 3,3’-diindolylmethane differentially regulate aryl hydrocarbon receptor and estrogen receptor alpha activity through multiple transcriptomic targets in MCF-7 human breast cancer cells
Authors: Siddhartha Das, Stine M. Ulven, Jason Matthews *
In general, the data in present studies are good and support the major conclusions of this manuscript. However, following issues need to be considered prior to considering the manuscript of publication.
[Major concerns]
1. MCF-7 cells: The authors claimed to have conducted their research on MCF-7 cells from the title of the paper, but additional explanation is needed regarding why MCF-7 cells were chosen among the numerous breast cancer cell types. Furthermore, there is a need for a discussion on whether MCF-7 cells are associated with unique genetic characteristics.
2. Abbreviations: The use of abbreviations when writing a paper has many advantages besides simplicity of expression. To use an abbreviation, first write the abbreviation in parentheses after the full name, and then use the abbreviation from Introduction to the final Conclusion. Abbreviations should only be used if they are repeatedly used and if they are not used again, only the full name should be used. In particular, because of the characteristics of IJMS, where Materials and Methods is arranged at the end of the paper, the original words and abbreviations are written in the order they are used from the introduction, and only when the abbreviation is used repeatedly, the abbreviation can be used until the conclusion. Examples: ARNT at Line 48 has already been abbreviated at Line 3; etc.
3. Materials and Methods section - When naming a particular chemical company, you must provide location information such as company name, city and/or state (abbreviation in the USA and Canada) and country. Once you have named a company with the information, you should only mention a company’s name thereafter. Information about several companies is wrong, so check and correct it. It is generally well written in this paper, but there are a few mistakes, so find them and correct them. Examples: ATCC at Line 408; Agilent at Line 439; etc.
4. Notation of Figure numbers in the manuscript: It would be advisable to place a period after "Fig" when referring to figure numbers in the main text of the paper, such as "Fig 1A-B" and etc.
[Minor concerns]
1. Line 400: 17b-estradiol should be written as 17β-estradiol.
2. Line 435: 10uM at Line 435 should be written as 10 μM.
3. Line 460: “Homo Sapiens” at Line 460 should be written as “Homo sapiens”.
4. Reference section: Author should consult and peruse carefully recent issues of the journal, International Journal of Molecular Sciences (IJMS), for format and style. Also double-check the abbreviations of journal names.
5. There is also several reference without page numbers and incomplete. Examples: 13, 16, 50, etc.
Overall, the manuscript can be considered to publication after minor revision as indicated above.

ijms-2607836
Title: Resveratrol and 3,3’-diindolylmethane differentially regulate aryl hydrocarbon receptor and estrogen receptor alpha activity through multiple transcriptomic targets in MCF-7 human breast cancer cells
Authors: Siddhartha Das, Stine M. Ulven, Jason Matthews *
In general, the data in present studies are good and support the major conclusions of this manuscript. However, following issues need to be considered prior to considering the manuscript of publication.
[Major concerns]
1. MCF-7 cells: The authors claimed to have conducted their research on MCF-7 cells from the title of the paper, but additional explanation is needed regarding why MCF-7 cells were chosen among the numerous breast cancer cell types. Furthermore, there is a need for a discussion on whether MCF-7 cells are associated with unique genetic characteristics.
2. Abbreviations: The use of abbreviations when writing a paper has many advantages besides simplicity of expression. To use an abbreviation, first write the abbreviation in parentheses after the full name, and then use the abbreviation from Introduction to the final Conclusion. Abbreviations should only be used if they are repeatedly used and if they are not used again, only the full name should be used. In particular, because of the characteristics of IJMS, where Materials and Methods is arranged at the end of the paper, the original words and abbreviations are written in the order they are used from the introduction, and only when the abbreviation is used repeatedly, the abbreviation can be used until the conclusion. Examples: ARNT at Line 48 has already been abbreviated at Line 3; etc.
3. Materials and Methods section - When naming a particular chemical company, you must provide location information such as company name, city and/or state (abbreviation in the USA and Canada) and country. Once you have named a company with the information, you should only mention a company’s name thereafter. Information about several companies is wrong, so check and correct it. It is generally well written in this paper, but there are a few mistakes, so find them and correct them. Examples: ATCC at Line 408; Agilent at Line 439; etc.
4. Notation of Figure numbers in the manuscript: It would be advisable to place a period after "Fig" when referring to figure numbers in the main text of the paper, such as "Fig 1A-B" and etc.
[Minor concerns]
1. Line 400: 17b-estradiol should be written as 17β-estradiol.
2. Line 435: 10uM at Line 435 should be written as 10 μM.
3. Line 460: “Homo Sapiens” at Line 460 should be written as “Homo sapiens”.
4. Reference section: Author should consult and peruse carefully recent issues of the journal, International Journal of Molecular Sciences (IJMS), for format and style. Also double-check the abbreviations of journal names.
5. There is also several reference without page numbers and incomplete. Examples: 13, 16, 50, etc.
Overall, the manuscript can be considered to publication after minor revision as indicated above.
Reviewer 2 Report
This a report by Das et al. that uses CHIP-seq and RNAseq techniques to study targeting and gene regulation induced by receptors (ERalpha and AhR) and their respective/co-respective ligands (RES, DIM, E2, TCDD, and in various combinations). The report is well-written with interesting results and approach. Some significant findings from this report are the cross-talk interactions between ERa/AhR ligands, and the predominance of ERa over AhR in some cases.
However, some issues were noted that might need to be addressed:
1. Please provide justification for the doses of ligands used (TCDD, RES, E2, DIM), and appropriate citations if this is means for justification. DIM was found to induce more AhR-related genes than TCDD, which is quite interesting since TCDD has been shown to have much higher affinity to AhR than DIM. Is this related to doses (DIM is significantly higher)?
2. An exciting conclusion is that ERa appears to dominate over AhR. Is this related to the cell line used (MCF-7)? The source is a female of course, so is ERa known to be dominate in female vs male? If so, what do these results implicate for male-derived cells (e.g. do you expect similar results)? This could be addressed in Discussion.
3. Please provide justification for timepoints used. While it is common to look at transcription binding at 1 hour and gene expression changes later (e.g 6-24 hours), it might be benificial to the reader to understand why authors selected these timepoints.
4. Table 1 - it appears Table 1 is labeled incorrectly based on the results highlighted in the report (e.g. DIM induces AHRE, not RES).
5. Line 288 - Why was the indicated data "not shown" in this particular case?
Minor Issues:
1. Line 279 - should CYP1A1/CYP1B1 be italized?
2. line 75 - awkward wording; should it be "although there are some studies WHICH report ERa inhibiting activity"
